



# Global cotton production under climate change - Implications for yield and water consumption

Yvonne Jans[1,2], Werner von Bloh[1], Sibyll Schaphoff[1], and Christoph Müller[1]

[1]Potsdam Institute for Climate Impact Research, Member of the Leibniz Association, P.O. Box 60 12 03, DE-14412 Potsdam, Germany
[2]Humboldt-Universität zu Berlin, Department of Geography, Unter den Linden 6, DE-10099 Berlin, Germany

**Correspondence:** Yvonne Jans (jans@pik-potsdam.de)

**Abstract.** Being an extensively produced natural fiber on earth, cotton is of importance for economies. Although the plant is broadly adapted to varying environments, growth and irrigation water demand of cotton may be challenged by future climate change. To study the impacts of climate change on cotton productivity in different regions across the world and the irrigation

water requirements related to it, we use the process-based, spatially detailed biosphere and hydrology model LPJmL. We find our modelled cotton yield levels in good agreement with reported values and simulated water consumption of cotton production similar to published estimates. Following the ISIMIP protocol, we employ an ensemble of five General Circulation Models under four Representative Concentration Pathways (RCPs) for the 2011–2099 period to simulate future cotton yields. We find that irrigated cotton production does not suffer from climate change if $CO_2$ effects are considered, whereas rainfed

production is more sensitive to varying climate conditions. Considering the overall effect of a changing climate and $CO_2$ fertilization, cotton production on current cropland steadily increases for most of the RCPs. Starting from ~65 million tonnes in 2010, cotton production for RCP4.5 and RCP6.0 equates to 83 and 92 million tonnes at the end of the century, respectively. Under RCP8.5, simulated global cotton production raises by more than 50% by 2099. Taking only climate change into account, projected cotton production considerably shrinks in most scenarios, by up to one-third or 43 million tonnes under RCP8.5. The

simulation of future virtual water content (VWC) of cotton grown under elevated $CO_2$ results for all scenarios in less VWC compared to ambient $CO_2$ conditions. Under RCP6.0 and RCP8.5, VWC is notably decreased by more than 2,000 $m^3t^{-1}$ in areas where cotton is produced under purely rainfed conditions. By 2040, the average global VWC for cotton declines in all scenarios from currently 3,300 to 3,000 $m^3t^{-1}$ and reduction continues by up to 30% in 2100 under RCP8.5. While the VWC decreases by the $CO_2$ effect, elevated temperature (and thus water stress) reverse the picture. Except for RCP2.6, the global

VWC of cotton increase slightly but steadily under the other RCPs until mid century. RCP8.5 results in an average global VWC of more than 5,000 $m^3t^{-1}$ by end of the simulation period. Given the economic relevance of cotton production, climate change poses an additional stress and deserves special attention. Changes in VWC and water demands for cotton production are of special importance, as cotton production is known for its intense water consumption that led, e.g., to the loss of most of the Aral sea. The implications of climate impacts on cotton production on the one hand, and the impact of cotton production

on water resources on the other hand illustrate the need to assess how future climate change may affect cotton production and





its resource requirements. The inclusion of cotton in LPJmL allows for various large-scale studies to assess impacts of climate change on hydrological factors and the implications for agricultural production and carbon sequestration.

# 1 Introduction

Being an extensively produced natural fiber on earth, cotton (*Gossypium Spp*) is providing income to millions of farmers.

According to the World Bank Atlas (Sheth, 2017), eight of the top-10 cotton-producing countries are classified as developing countries and their exports of the crop reached ∼ US$ 30 billion in 2017 (ITC, 2019) (full overview in Table S1). Particularly in the West African region – the world's third largest cotton exporter (following North America and Central Asia) – cotton has played an important part in the economic development and has remained a key source of livelihood for many farmers (Hussein et al., 2005; Perret and Bossard, 2006). Worldwide, cotton is already broadly adapted to growing in temperate, subtropical and

tropical environments, but growth may be challenged by future climate change (Bange et al., 2016). Climate change is likely to affect cotton production both positive and negative. Temperature influences cotton growth and development by determining rates of fruit production, photosynthesis and respiration (Turner et al., 1986; Hearn and Constable, 1984).

However, the growth of cotton plants differs at varying stages of plant development and by plant organ (Burke and Wanjura, 2010) and thus a temperature optimum for cotton cannot be defined. Yield and growth of cotton are directly affected by a high

temperature. Additionally, hot weather conditions increase the evaporative demand in cotton plants leading to more intense water stress (Hall, 2000). Elevated atmospheric carbon dioxide concentrations ([$CO_2$]) on the other hand are expected to increase cotton yields as cotton is a $C_3$ crop (Kimball, 2016). Numerous FACE studies demonstrated a strong reaction of cotton yield and growth to an increased $CO_2$ concentration (Kimball, 1983; Cure and Acock, 1986; Hileman et al., 1994; Hendrix et al., 1994; Reddy et al., 1997; Mauney et al., 1994; Bhattacharya et al., 1994). Likewise, water-use efficiency (WUE) can be

improved by $CO_2$ enrichment because it increases biomass and causes partial stomatal closure at the same time, consequently reducing transpiration (Mauney et al., 1994; Hileman et al., 1994; Broughton, 2015; Ko and Piccinni, 2009).

Crop models have been used to assess the effect of changing climate conditions on crop productivity but the main focus has been on major staple crops, such as maize, wheat, rice and soybean (e.g. Challinor et al., 2014; Rosenzweig et al., 2014; Müller et al., 2015; Pugh et al., 2016; Schleussner et al., 2018) that provide the majority of calories to human nutrition (Yahia et al.,

2019; Welch and Graham, 2004).

The response of other crops has been assessed less thoroughly, despite their importance for economies or human nutrition. With this study, we aim to examine the impacts of climate change on cotton productivity in different regions across the world. We therefore add cotton as an additional crop to the global dynamic vegetation, hydrology and crop growth model LPJmL version 4.0 (Schaphoff et al., 2018b). We provide an evaluation of model skill by comparing simulated cotton yields to yield

statistics (FAO et al., 2018). To study climate change impacts on future cotton productivity, we simulate future cotton yields and related irrigation water requirements under a set of future climate scenarios (Hempel et al., 2013), following the ISIMIP protocol (Warszawski et al., 2014).





## 2 Materials and Methods

### 2.1 The LPJmL model

The global dynamic vegetation model LPJmL is a well-established and thoroughly evaluated model (Schaphoff et al., 2018b, a; Müller et al., 2017) that is unique in combining natural vegetation, hydrology and managed ecosystems (croplands, pastures) in one consistent framework for gridded large-scale applications. The model has been extensively described by Schaphoff et al. (2018b) and we here only provide a short summary of the most relevant features for this study and the extensions implemented for cotton. Agricultural crops have been implemented as annual crops with daily computation of photosynthesis, autotrophic

respiration, evapotranspiration and allocation of assimilates to plant organs (Bondeau et al., 2007). Individual crops are grown on separate spatial units (stands) within each grid cell so that different crops do not compete for water and light, mimicking monocultures. Also purely rainfed and irrigated crop cultivation can be simulated on separate stands and irrigation water can be applied by different irrigation techniques and can be limited by actual freshwater availability (Jägermeyr et al., 2015). LPJmL – here operated at a 0.5 arc-degree (°) spatial resolution – simulates processes underlying the growth and productivity of both

natural and agricultural vegetation (Sitch et al., 2003; Bondeau et al., 2007; Lapola et al., 2010; Rolinski et al., 2017). The model represents 10 plant functional types (PFT) as well as 12 crop functional types (CFT) and three bioenergy plantation systems (Beringer et al., 2011). In LPJmL, carbon, water and energy fluxes are closely linked to reproduce plant growth dynamics and to account for the effects of changes in climate conditions and water availability (Gerten et al., 2004, 2007). Several features such as river routing (Rost et al., 2008a), irrigation systems (Jägermeyr et al., 2015), a soil hydrological and

carbon distribution scheme (Schaphoff et al., 2013), and a fire module (Thonicke et al., 2010) further improved the model. The calculation of photosynthesis is based on the Farquhar model (Collatz et al., 1991; Haxeltine and Prentice, 1996; Sitch et al., 2003). Water consumption is ruled by plant physiology and the coupling between vegetation and water cycle enables the separation of productive (transpiration) and unproductive (interception, evaporation) portions of plant water use. Moreover, water flows are divided into green (precipitation) and blue (irrigation) water (Rost et al., 2008b; Jägermeyr et al., 2015, 2016).

The evaluation of various model components, e.g. crop yields, evapotranspiration and river discharge has shown that LPJmL is a tool suitable to analyse changes in vegetation and water. Schaphoff et al. (2018a) provide a comprehensive evaluation of the LPJmL model.

### 2.2 Implementation and parameterisation of cotton

Twelve crop types are already implemented in LPJmL (temperate and tropical cereals, pulses, maize, rice, temperate and
tropical roots, sunflower, soybeans, groundnut, rapeseed, sugar cane) (Bondeau et al., 2007; Lapola et al., 2010). In this study we include cotton, which was originally implemented as a perennial crop in LPJmL by Fader et al. (2015) for the Mediterranean region. However, in most parts of the world, cotton is cultivated as an annual crop (Ritchie et al., 2007; Whitaker et al., 2018). We modify the modeling approach developed by Fader et al. (2015) accordingly and implement cotton into LPJmL version 4.0 (Schaphoff et al., 2018b) as an annual crop. Similar to Fader et al. (2015), the cotton plant is simulated and parametrised as



an agriculural tree and we calculate phenology and growth on a daily basis (see below). We adopt most of the key parameters used by Fader et al. (2015) and adjust values for plant density and temperature optimum for photosynthesis (Table 1).

**Table 1.** Key parameters of cotton according to Fader et al. (2015). Values marked with an asterisk (*) were adjusted. **K$_{est}$**: tree density range; **HR**: harvest ratio; **T$_b$**: base temperature; **Ph$_{opt}$**: optimum temperature range for photosynthesis; **T$_{lim}$**: lower and upper coldest monthly mean temperature; **WCF**: conversion factor (moisture content) from dry to fresh matter.

| Crop | Seasonality | K$_{est}$ (trees ha$^{-1}$) | HR (frac) | T$_b$ (° C) | Ph$_{opt}$ (° C) | T$_{lim}$ (° C) | WCF (% of DM) |
|---|---|---|---|---|---|---|---|
| Cotton | Deciduous broadleaved | 30,000 - 100,000* | 0.19 | 15 | 16 to 32* | -10 to 40 | 91 |

### 2.2.1 Phenology and growth

Wild cotton is a deciduous perennial tree and the fruiting habit of the plant is not clearly established, i.e. vegetative and reproductive growth occur at the same time (Ritchie et al., 2007). During the growing period the leaves supply photosynthates
to plant growth and the developing fruit and are shed only when the plant is stressed such as during drought, disease, nutrient starvation or frost (Wullschleger and Oosterhuis, 1990). The perennial nature of cotton, even its modern cultivars, is not helpful in achieving high yields of cotton lint and seed. Consequently, through breeding and changes in cultivation practices, cotton is now farmed as an annual crop to prevent diseases and optimize cotton production (Ritchie et al., 2007; Whitaker et al., 2018). Once the entire crop is mature, the leaves serve no useful purpose and their removal can be beneficial for mechanical
harvesting. Crop maturity is characterized by slowed development of new main-stem nodes, causing first-position white flowers to appear progressively closer the plant apex (Oosterhuis et al., 1992). To account for the current production system, cotton is implemented in LPJmL as small agricultural trees that are planted annually and removed at the end of the growing period, representing the annual production mode. The saplings are initialized with 2.3 gC of sapwood and a leaf-area index (LAI) of 1.6 m$^2$ m$^{-2}$. Phenology, gross primary production (*GPP*) and net assimilation (*NPP*) of cotton plants are calculated daily.
Fruit growth is expressed as daily carbon accumulation ($C_{fruit}$) of a fraction of *NPP* after square set, i.e. as soon as squares (pre-bloom fruiting buds) emerge. Square form at the initiation of a fruiting branch. The model implementation assumes that cotton fruit growth occurs after the fractional cover of green leaves has reached 60 % of full leaf cover, i.e. the phenology scaler *phen*=0.6. This follows the description of Ritchie et al. (2007) on the canopy and fruit development of cotton plants.

$$C_{fruit} = max(0, NPP) x HR \tag{1}$$

where *HR* is the harvest ratio and *NPP* the daily net primary productivity of the tree. On days with negative *NPP*, fruit growth is halted but accumulated yield is not reduced, reflecting that boll development dominates plant growth at this stage of reproductive growth (Ritchie et al., 2007). At the end of the growing period, cotton harvest *H* is determined as

$$H = \sum_{D_S}^{D_H} C_{fruit} \tag{2}$$





where the day of square set ($D_S$) and harvest day ($D_H$) define the length of the simulated reproductive period.

A possible simultaneous establishment of herbaceous PFTs in the same areas of agricultural trees, representing grasses and weeds (for modelling details see Schaphoff et al. (2018b)), can be simulated by LPJmL, but was turned off in the simulations here. This mimics effective weed control, mainly practiced in cotton farming today to reduce competition for water and nutrients (Ritchie et al., 2007; Whitaker et al., 2018).

### 2.2.2    Specification of planting densities, sowing dates and irrigation

A country-specific planting density ($k_{est}$) is used as model input, which is, apart from irrigation and sowing dates, the only management aspect that is explicitly considered. These country-specific planting densities have been taken from literature (Abdullaev et al., 2007; Iqbal et al., 2012; Venugopalan et al., 2013; Dong et al., 2006; Bednarz et al., 2006; Zhi et al., 2016; Khan et al., 2017; Bozbek et al., 2006; Echer and Rosolem, 2015; Dai and Dong, 2014; Vaughan, 2005; Akhtar et al., 2002) and are shown in Figure S1. The cotton specific planting density parameter (plants $\mathrm{m^{-2}\,a^{-1}}$) was introduced similar to the

annual establishment rate $k_{est}$ of woody PFT individuals (Schaphoff et al., 2018b).

     The growing season of cotton plants is prescribed from the sources specified in section *Modeling protocol and input data*. Thereby the sowing date defines the start of the growing period and ranges between Julian day 1 and 335 of the year of sowing. The prescribed growing season length varies from 153 to 243 days of year for cotton plants to reach harvest.

     Five hydrologically and thermally active layers represent the soil profile in LPJmL where roots access water, depending

on their PFT/CFT-specific root distribution (Schaphoff et al., 2018b). The soil water content of the first layer determines the infiltration rate and water not infiltrated forms the surface runoff. Similar to the infitration approach, the percolation rate is limited by soil moisture of the lower layer and excess water above saturation feeds the lateral runoff (Schaphoff et al., 2013).

     Cotton is produced in rainfed or irrigated systems, whereat irrigation generally serves to reduce the impacts of rainfall deficits and thus reduces interannual yield variability. However, the actual amount of water applied to fields is unknown and

determined by water availability, water management systems and economic rationale. The extent to which rainfall deficits are compensated by irrigation is thus not only a question of equipment for irrigation (Portmann et al., 2010) and water stress can still affect interannual yield variability in irrigated systems. In order to test the importance of deficit irrigation for cotton production, we performed several runs varying the fraction of soil pore space filled up in individual irrigation events from 0 (corresponding to purely rainfed conditions) to 1 (meets full irrigation demand) by increments of 0.25.

If the soil water content in the upper 50 cm of the soil falls below 90% of field capacity an irrigation event is triggered. Soil water lower than atmospheric water demand requires a daily net irrigation (NIR; mm). NIR is calculated as the amount of water needed to fill soil water up to field capacity $W_{fc}$ in that upper root layers of the soil.

$$NIR = max(0, (W_{fc} - w_a)) \tag{3}$$

where $w_a$ is the soil water in millimeters actual available .





Inefficiencies of different irrigation systems cause additional water needs to meet crop water demand. For that reason, LPJmL considers conveyance efficiency and calculates application requirements (*AR*) for each system. Consequently, the gross irrigation requirement (*GIR*; mm), i.e. the amount of water requested for abstraction results in

$$GIR = \frac{NIR + AR - Store}{E_c} \qquad (4)$$

    where *Store* is a storage buffer. The storage buffer is filled up with available irrigation water not used due to available
precipitation and is released in the next irrigation event. A detailed explanation about the computation of *NIR*, *GIR*, and *Store* in LPJmL is given in Rost et al. (2008b) and Jägermeyr et al. (2015). The application requirements *AR* are calculated as

$$AR = max(0, (W_{sat} - W_{fc})xDU - w_{fw} \qquad (5)$$

    where $W_{sat}$ is the soil moisture content at saturation level ( in mm); *DU* is an irrigation system-specific scalar (no unit), to distribute irrigation water uniformly across the field, and $w_{fw}$ is the available free water (in mm) (Jägermeyr et al., 2015). Note
that the computation of *GIR* is relevant for simulations, in which irrigation water is constrained by available river discharge and reservoir capacity (e.g. Jägermeyr et al., 2015), but not here, where we assume explicit levels of deficit irrigation but no additional constraints on water limitation.

## 2.3   Modeling protocol and input data

All simulations are conducted at a spatial grid of 0.5° longitude/latitude resolution with daily weather input data and annual data
on atmospheric carbon dioxide concentrations ([$CO_2$]). To simulate historical results, we ran the LPJmL model for the period 1901 – 2011 using the Climate Research Unit's (CRU) TS 3.23 monthly data for temperature, wet days and cloudiness (Harris et al., 2014), and precipitation data (Version 5) provided by the Global Precipitation Climatology Centre (GPCC) (Rudolf et al., 2011). Monthly weather input data is converted to daily data, using an internal weather generator (Schaphoff et al., 2018b). Data on [$CO_2$] refers to records at the Mauna Loa station (Tans and Keeling, 2015). A 120-year spin-up (recycling the first
30 years of input climatology) preceded transient runs to bring water fluxes and soil temperatures into dynamic equilibria. As soil carbon pools have no effect on cotton productivity in this version of the model, a longer spinup to correctly initialize soil and vegetation carbon pools (Schaphoff et al., 2018b) is not necessary here. Simulations for future periods are conducted for four different Representative Concentration Pathways (RCPs), RCP2.6, RCP4.5, RCP6.0 and RCP8.5 (Moss et al., 2010), each implemented by five different General Circulation Models (GCMs): GFDL-ESM2M (Dunne et al., 2012, 2013), HadGEM2-ES
(Jones et al., 2011), IPSLCM5A-LR (Dufresne et al., 2013), MIROC-ESM (Watanabe et al., 2011), and NorESM1-M (Bentsen et al., 2013) that have been bias-corrected as described by Hempel et al. (2013). Data on [$CO_2$] for future periods is taken from the corresponding RCP data sets (Moss et al., 2010) as provided by the ISIMIP project (Frieler et al., 2017). To assess how cotton plants respond to future climate change we ran the model for the time span 1951 – 2099, again preceded by a 120-year spin-up period. The averaging time for historical yields differs (depending on the purpose) and is indicated for each





figure. Future yield projections are presented as 2090 – 2099 averages or as full time series. The spatial distribution of cropland
dedicated to cotton was taken from the land-use data set MIRCA2000 (Portmann et al., 2010) which provides both rainfed and
irrigated harvested areas around the year 2000 with a spacial resolution of 5 arc min (Figure S2).

Sowing dates and growing period were provided as gridded model input, combining sowing and harvest information provided
by the ICACs World Cotton Calendar (WCC) (Committee, 2014) and Portmann et al. (2010). More precisely, we used the WCC

data and filled gaps with data offered by MRICA (Figure S3). In this study, simulated cotton yields should be understood as
the entire cotton fruit, that is both cotton lint and cottonseed. For comparison we used observations published by the FAO et al.
(2018) reported as "Seed cotton" there.

## 3  Results

### 3.1  Evaluation of model performance

In order to evaluate the performance of this extended LPJmL model version, simulated historical cotton yields are compared
to observed data (Figure 1) published by FAOstat (FAO et al., 2018). The modelled cotton yield levels are in good agreement
with reported values. Statistical analysis for both, the top-10 cotton-producing countries and for cotton-growing countries in
West Africa show that simulated national yield levels can well reproduce reported national yield levels (Figure 1). For the
top-10 cotton-producing countries, cotton yields simulated under full irrigation often match FAO values best. The whiskers

that depict the range of yield levels simulated with different irrigation options on the irrigated cotton cropland often reach
the zero line, indicating that cotton production in these countries (Pakistan, Turkmenistan, Turkey, Uzbekistan etc.) is not
possible without irrigation. National yield levels can also be reproduced well in West Africa, where in contrast to the top-10
producer countries hardly any irrigated cotton production exists (Figure S2). An overview of the model performance for all
cotton-growing countries is given in the supplementary information, Table S2.

For these simulations, the planting densities in LPJmL have not been calibrated against observed yield levels, but are based
on reported planting densities (Figure S1).

The model simulations can reproduce statistically significant shares of reported variability in time of intensely-managed
top-producing countries, such as the USA and Australia as well as a few West African countries (Figures 2 and S4) and other
countries (Table S2).

The model also reproduces some of the historical interannual variation in global cotton production (Figure 3).

The spatial pattern of cotton yields is shown in Figure S5.

We further evaluate the model results with respect to the water consumption of cotton production against figures provided
by (Chapagain et al., 2006), averaged for the time period 1997-2001. For reasons of comparability, we therefore followed the
concept of "virtual water content" (Allan, 1997, 1998) and calculated the virtual water content of cotton ($\text{t m}^{-3}$) as the ratio

of the water (green and blue in $\text{m}^3 \text{ha}^{-1}$) consumed during the entire period of crop growth to the corresponding crop yield
($\text{t ha}^{-1}$). We find that LPJmL simulations of water consumption of cotton production are in good agreement with the estimates
of (Chapagain et al., 2006) with respect to the order of magnitude and spatial variability (Figure S6 and Table 2). The virtual





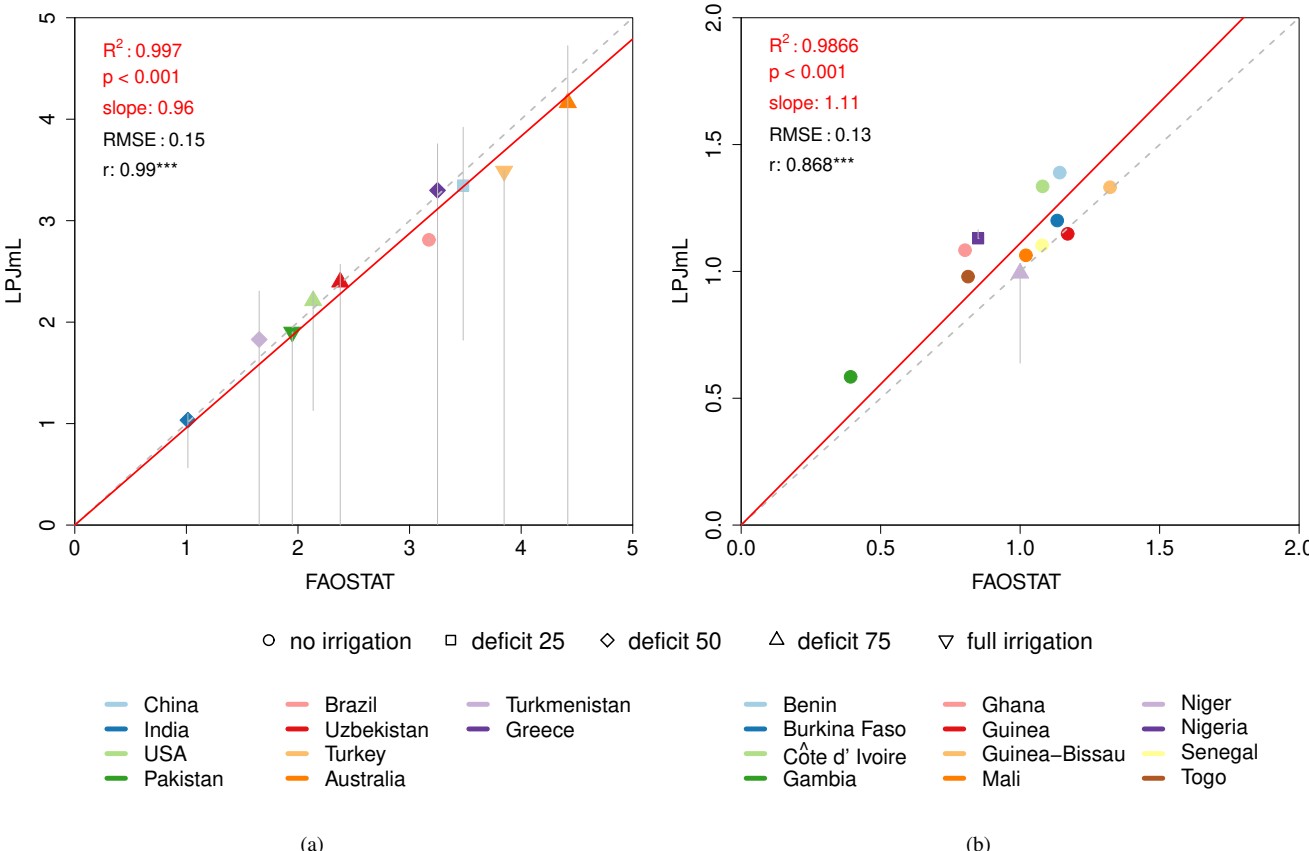

**Figure 1.** Comparing simulated cotton yields [t ha$^{-1}$] to observed values for (a) the top-10 cotton producing countries and (b) West African countries. Whiskers indicate the yield range of different irrigation options on irrigated cotton cropland in these countries (Portmann et al., 2010). LPJmL yield data and FAOSTAT yield data were both averaged over the time period 2000-2009.

water content is quite variable across regions and mainly in an inverse relation of the yield pattern (Figure S5), suggesting that spatial yield variability is higher than the spatial variability in actual evapotranspiration (AET). As virtual water content is a

criterion for water-use efficiency (Hoekstra, 2003; Hoekstra and Mekonnen, 2012; Zhuo and Hoekstra, 2017), moderate values in regions with high irrigation shares (compare Figures S6 and S7) point to an efficient use of (blue) water. The efficiency of blue-water use depends on management practices, such as irrigation techniques, irrigation strategies, and mulching practices (Gleick, 2003; Perry, 2007; Perry et al., 2009; Zhuo and Hoekstra, 2017) to reduce non-beneficial losses (soil evaporation) as well as on other yield-reducing factors, such as nutrient limitations or pests. In the Indo-Gangetic plain, drip irrigation of cotton

is only applied in experimental fields and farmers grow cotton by applying irrigation water through flood irrigation (Thind et al., 2008; Aujla et al., 2008). Here, the water consumption of cotton production is at the high end, indicating substantial non-beneficial water losses (Thind et al., 2010).







**Figure 2.** Comparing interannual yield variability for the top-10 cotton producing countries. Numbers in each plot depict the correlation coefficient between simulated residuals and FAOSTAT residual data. The different irrigation options (on irrigated cropland only) are shown in coloured lines. The colour of the correlation coefficient indicates to best fitting irrigation option. For Turkmenistan, yields have only been reported from 1992 onward (FAO et al., 2018), so only these years are shown in the plot.





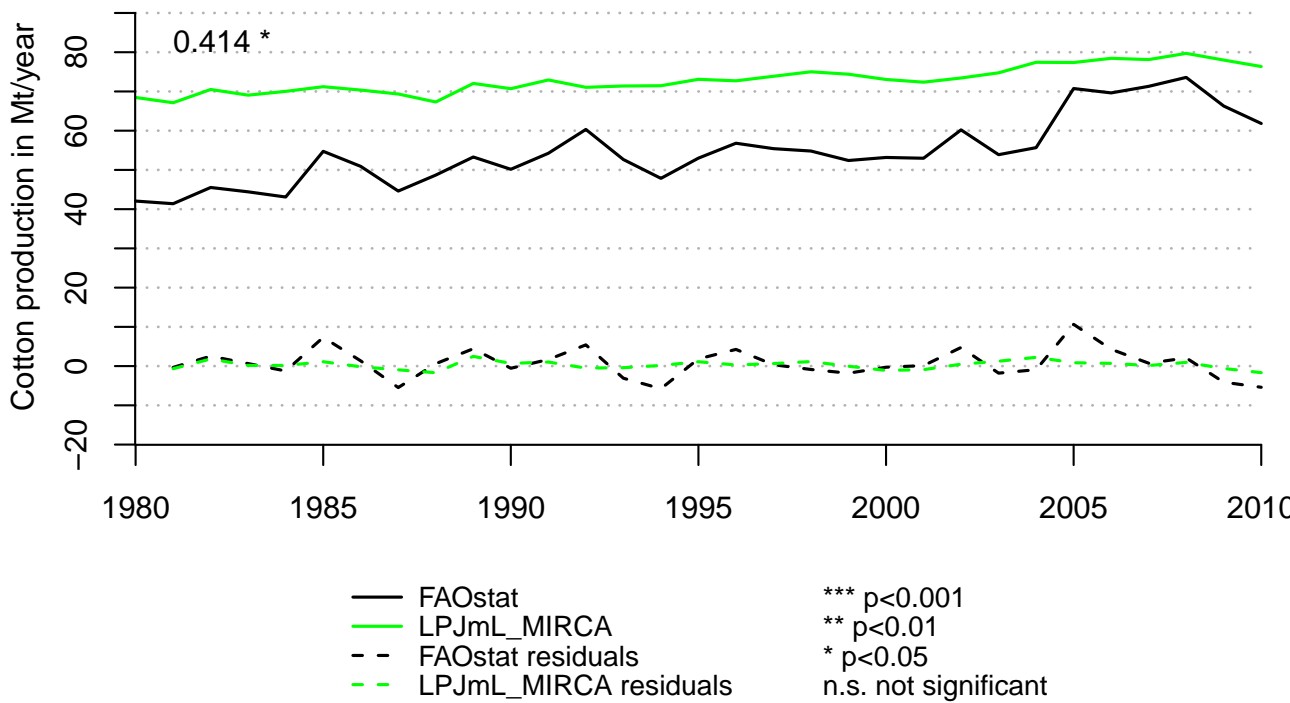

**Figure 3.** Time series of historical global cotton production [million tonnes/year]. The number in the plot depicts the correlation coefficient between simulated residuals and FAOSTAT residuals.





**Table 2.** Virtual water content and consumptive water use for cotton production in the major cotton producing countries for the period 1997-2001. Reference data taken from Chapagain et al. (2006), here referred to as *"C06"*. $VWC_{blue}$: blue virtual water content, $VWC_{total}$: total virtual water content, $A_{irrig}$: irrigated harvested cotton area, $A_{total}$: total harvested cotton area.

| | Virtual water content [m$^3$/ton] | | | | $VWC_{blue}$/$VWC_{total}$ | | $A_{irrig}$/$A_{total}$ | | Consumptive water use [mm] | | | |
| | Total | | Blue | | | | | | Total | | Blue | |
| | LPJmL | C06 | LPJmL | C06 | LPJmL | C06 | LPJmL | C06 | LPJmL | C06 | LPJmL | C06 |
|---|---|---|---|---|---|---|---|---|---|---|---|---|
| Argentina | 2,564 | 7,700 | 142 | 2,307 | 0.06 | 0.30 | 0.08 | 1 | 775 | 877 | 43 | 263 |
| Australia | 2,536 | 2,278 | 1,677 | 1,408 | 0.66 | 0.62 | 1 | 0.9 | 1,177 | 843 | 778 | 521 |
| Brazil | 3,235 | 2,621 | 20 | 46 | 0.01 | 0.02 | 0.02 | 0.15 | 895 | 551 | 5 | 10 |
| China | 1,907 | 2,018 | 481 | 760 | 0.25 | 0.38 | 0.41 | 0.75 | 716 | 638 | 180 | 240 |
| Egypt | 2,727 | 4,231 | 2,605 | 4,231 | 0.96 | 1.00 | 1 | 1 | 963 | 1,009 | 921 | 1,009 |
| Greece | 2,554 | 2,338 | 1,586 | 1,808 | 0.62 | 0.77 | 1 | 1 | 937 | 707 | 582 | 547 |
| India | 5,232 | 8,662 | 1,773 | 2,150 | 0.34 | 0.25 | 0.36 | 0.33 | 563 | 538 | 191 | 134 |
| Mali | 5,738 | 5,218 | 0 | 1,468 | 0.00 | 0.28 | 0 | 0.25 | 594 | 538 | 0 | 151 |
| Mexico | 4,191 | 2,508 | 2,538 | 1,655 | 0.61 | 0.66 | 0.97 | 0.95 | 933 | 746 | 565 | 492 |
| Pakistan | 5,486 | 4,914 | 4,377 | 3,860 | 0.80 | 0.79 | 1 | 1 | 1,033 | 850 | 824 | 668 |
| Syria | 2,689 | 3,339 | 2,005 | 3,252 | 0.75 | 0.97 | 1 | 1 | 1,008 | 1,309 | 751 | 1,275 |
| Turkey | 2,779 | 3,100 | 1,796 | 2,812 | 0.65 | 0.91 | 1 | 1 | 943 | 963 | 609 | 874 |
| Turkmenistan | 4,076 | 6,010 | 3,538 | 5,602 | 0.87 | 0.93 | 0.98 | 1 | 926 | 1,025 | 804 | 956 |
| USA | 3,461 | 2,249 | 966 | 576 | 0.28 | 0.26 | 0.37 | 0.52 | 775 | 419 | 216 | 107 |
| Uzbekistan | 3,616 | 4,460 | 2,854 | 4,377 | 0.79 | 0.98 | 1 | 1 | 911 | 999 | 719 | 981 |
| Global average | 3,338 | 3,644 | 1,397 | 1,818 | 0.42 | 0.50 | 0.491 | – | 755 | – | 320 | – |

For the evaluation of the modeled cotton yield response to elevated [$CO_2$], we compare simulated yield effects to those reported from Open-Top Chamber (OTC) and Free Air Carbon dioxide Enrichment (FACE) experiments. Kimball (2016) report strong yield increases in cotton bolls under elevated [$CO_2$] (~38 %), which is a stronger yield response than most other crops. Experimental data from Kimball et al. (1992) and Mauney et al. (1994) also show that the level of water and nutrient availability affects the relative cotton yield response to elevated [$CO_2$]. Similarly, LPJmL yields also result in a strong response depending on the level of [$CO_2$] increase and water stress (see Figure S8). Observational data are only available for one OTC site (Phoenix, AZ, USA) and one FACE site (Maricopa, AZ, USA) so that it remains unclear how the cotton yield response to





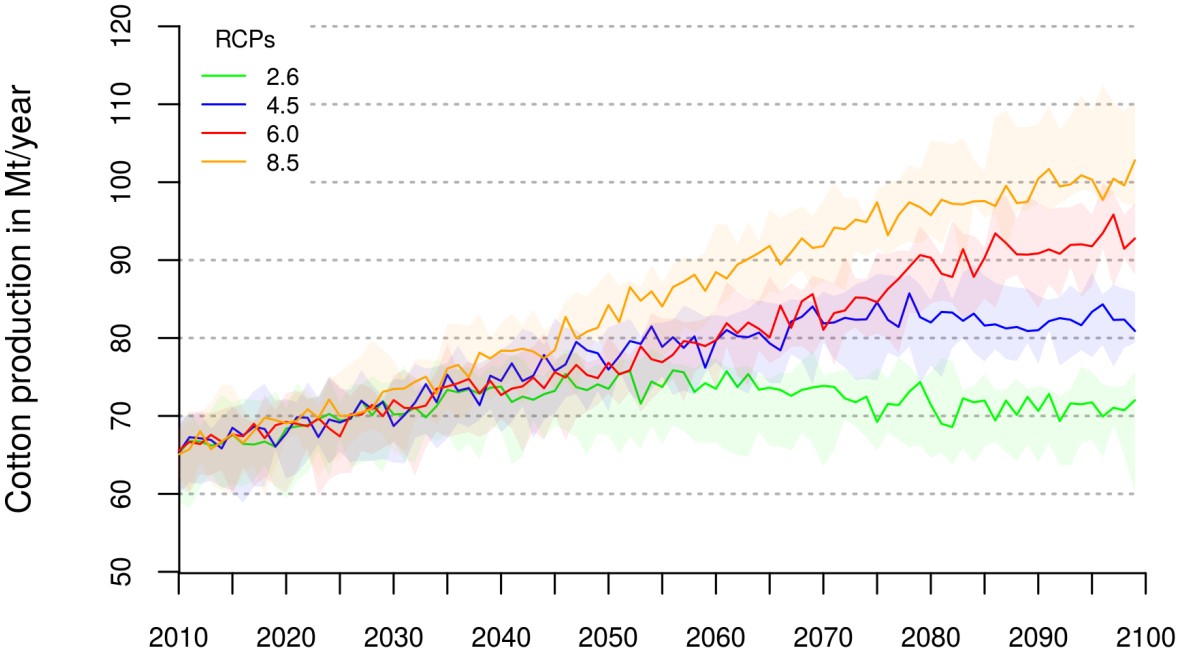

**Figure 4.** Simulated global cotton production [million tonnes] for different RCPs. Transparent colours show the uncertainty ranges of 5 different GCM patterns.

elevated [$CO_2$] varies across different climate zones and management regimes. However, the range of simulated yield increases under elevated [$CO_2$] seems to be often adequately reproduced by LPJmL in the corresponding grid cells (Figure S8).

### 3.2   Climate change impacts on cotton production

Considering the overall effect of climate change and $CO_2$ fertilization, future cotton productivity slightly increases – starting from ~65 million tonnes in 2010 – until 2040 for all RCPs similarly by (~10 %). For RCP2.6, global cotton production on

current cropland slightly declines after 2040, while for the remaining RCPs simulated production on current cropland steadily increases. Looking at at RCP4.5 and RCP6.0, cotton production equates to 83 and 92 million tonnes at the end of the century, respectively. Under RCP8.5, simulated global cotton production raises by more than 50 % up to 102 million tonnes by 2099 (Figure 4 and Figure S9 for relative changes).

     Spatial patterns of projected changes in cotton yields (Figure S10) show that increases are mainly expected in cooler or

irrigated environments (Figure S2), but exceptions exist, such as Pakistan and northern India, where cotton yields are project to only slightly increase despite irrigation. Overall, the spatial patterns of projected yield increases seem to be quite static but scale with the emission scenario (RCP; Figure S10).

     However, taking only climate change (temperature, precipitation, radiation) into account, i.e. ignoring yield stimulation from elevated [$CO_2$], projected global cotton production considerably shrinks in most scenarios. While projected cotton production





for RCP2.6 hardly changes over the entire simulation period, global cotton production is reduced by about 6 and 10 million tonnes (10% and 15%) by the end of the 21$^{st}$ century under RCP4.5 and RCP6.0, respectively. Under RCP8.5, global cotton production on current cropland decreases by end of the century by more than one-third or 43 million tonnes compared to the current production level, if no $CO_2$ fertilization effect is assumed (Figure S11 and Figure S12 for relative changes).

   Without the beneficial effect of elevated [$CO_2$], climate change leads to yield declines in most of the current cotton pro-

duction area (Figure S13). Across all RCPs the spatial variation of impacts shows a diverse pattern, resulting in climate-only induced yield losses up to $2\,\text{t}\,\text{ha}^{-1}$) in large parts of the cotton production area. Projected cotton yields for the high-end emission scenario (RCP8.5) decline significantly in Bolivia, Argentina, Iraq, Syria, and Egypt. Considerable losses are also projected for the USA, Brazil, India, Pakistan, Central Asia, the southeastern part of China, and Australia if no effects of $CO_2$ fertilization are assumed. Only for Peru, Northeast-China and some parts of Central Asia, the simulations project sustained cotton yield

gains under climate change only.

### 3.3    Climate change impacts on irrigation water consumption

While changes in atmospheric $CO_2$ may turn into enhanced water use or water-use efficiency of cotton production, the impact of elevated $CO_2$ on cotton growth depends also on plant water availability. The simulation of future virtual water content of cotton grown under elevated $CO_2$ results for all scenarios in less virtual water content compared to ambient $CO_2$ conditions

(Figure 5). While there is only a slight decrease in virtual water content of cotton under RCP2.6, this effect is continuously strengthened across the emission scenarios (RCPs) for both well-watered and water-stressed conditions. Under RCP6.0 and RCP8.5, virtual water content is notably decreased by more than $2{,}000\,\text{m}^3\text{t}^{-1}$ in areas where cotton is produced under purely rainfed conditions, e. g. in West Africa and India. By 2040, the average global virtual water content for cotton declines in all scenarios from currently 3,300 to $3{,}000\,\text{m}^3\text{t}^{-1}$ (Figure S14(a)). Thereafter it slightly increases again under RCP2.6 while

reduction continues for the remaining scenarios until end of the century. The most considerable decrease by 30% results in 2100 under RCP8.5 (Figure S14(b)).

   While the virtual water content improves by the $CO_2$ effect, elevated temperature (and water stress) reverse the picture. Except for RCP2.6, the global virtual water content of cotton increase slightly but steadily under RCPs until mid century. For RCP4.5 and RCP6.0 this development continues resulting in a virtual water content roughly 10% above the current value by

2100 (Figure S15(b)) if no $CO_2$ effect is assumed. The most obvious alteration is projected for RCP8.5 where the changing climate without accounting for the $CO_2$ effect leads to an average global virtual water content of more than $5{,}000\,\text{m}^3\text{t}^{-1}$ by end of the simulation period (Figure S15(a)). The spatial pattern reveals that in all scenarios the virtual water content increases in all regions if $CO_2$ effects are not accounted for. However, most drastic changes occur again in West Africa and India, where also the strongest changes are observed when $CO_2$ effects are accounted for. While the $CO_2$ effect leads to decreasing virtual

water content in these regions, it increases by $2{,}000\,\text{m}^3\text{t}^{-1}$ if the $CO_2$ effects are not accounted for (Figure S16).

**Figure 5.** Simulated changes in virtual water content of seed cotton [m³t⁻¹] for different RCPs. The spatial pattern of rainfed and irrigated cotton harvested areas was kept constant at the pattern of the year 2000 as provided by Portmann et al. (2010). Gray indicates areas currently not used for cotton production. Yields were averaged over the 5 GCM patterns and over the period 2090-2099.





## 4  Discussion

### 4.1  Model performance

The model can reproduce national yield levels very well (Figure 1). This can in part be expected as we use the best performing level of deficit irrigation in the comparison as well as national planting densities and reported growing seasons. However,

overall yield levels are often not very sensitive to smaller changes in irrigation levels (*full* vs. *deficit 75*) and it is plausible to assume that irrigation typically is applied in quantities that are sufficient to eliminate the majority of water stress. Also, yield levels in countries with no or little irrigation can also be well reproduced. National planting densities have been taken from literature sources and were not selected to match observed yield levels. A wide range of (national) cotton planting densities is reported in the literature, which would allow for further modification of this parameter to refine our results. However,

field research has shown different effects of increasing plant density on cotton yield and understanding how cotton growth is affected by that parameter multiple interacting factors must be considered (Heitholt and Sassenrath-Cole, 2010). In this study, we therefore have selected planting densities corresponding to the lower end of the spread reported and kept these values static, but literature suggests that planting densities have changed over time, explaining part of the temporal variation in cotton yields (Venugopalan et al., 2013).

The temporal variation in cotton yields can only partly be reproduced. This comparison is hampered by using static management assumptions in the absence of good spatially and temporally resolved management data, which is a general difficulty in evaluating gridded crop models' performance (Müller et al., 2017). The contribution of weather variability on yield levels remains unclear as the yield variability in reported yield statistics is not only affected by variability in weather, but also by varying management conditions (Schauberger et al., 2016). Ray et al. (2012) reported that only 30 % of global yield variability

can be attributed to weather drivers for maize, wheat, rice and soybean and also Müller et al. (2017) found better agreement between crop model simulations and yield statistics in high-input countries, suggesting that agreement between crop model simulations and yield statistics can only be expected in countries where management conditions are stable. Larger jumps in yield statistics as e.g. in Brazil, China, India, Pakistan and several West-African countries suggest changes in management that cannot be expected to be reproduced by this modeling setup with static management assumptions (Dong et al., 2005; Venu-

gopalan et al., 2013; Rossi et al., 2004). Additionally, inconsistencies between different data sets used to determine agricultural areas dedicated to cotton (details in Fader et al. (2015)) cause significant deviations from the annual harvested cotton areas provided by FAO et al. (2018), which was also reported as a source of uncertainty for other crops (Porwollik et al., 2017).

The good agreement with Chapagain et al. (2006) on virtual water content of cotton production adds further trust to the model simulations, as productivity and water consumption are intrinsically coupled in the model. Even though the estimates

from Chapagain et al. (2006) are also model-based estimates rather than observations, the simulated patterns are plausible and have been achieved with different methods.





## 4.2 Implications of climate impacts

Elevated $[CO_2]$ has been shown to increase leaf photosynthetic rates and crop radiation-use efficiency (Hileman et al., 1994; Idso and Idso, 1994; Reddy and Zhao, 2005; Broughton, 2015) and reduce transpiration at the leaf level through reduced

stomatal conductance (Hileman et al., 1994; Broughton, 2015; Zhao et al., 2004), in cotton. Both effects potentially lead to improvements in growth and yield. A broad range of yield increases, averaging around 38% has been reported for cotton bolls for an increase in $[CO_2]$ of 190 ppm (360 to 550 ppm), which is substantially stronger than the average response in most other crops, but only based on a very small set of experiments (Kimball, 2016). In line with changes in transpiration rates for canopies under elevated $[CO_2]$, Mauney et al. (1994) reported increased water-use efficiency as a function of increasing

biomass production rather than a reduction in water use in the FACE experiments. In contrast, Reddy et al. (2005b) showed that increase in temperatures above optimum decrease cotton yield due to increased boll abscission and smaller boll size. In their experiment, even a significant increase in $[CO_2]$ did not fully compensate the negative effects on yields. The authors conclude that future increases in $[CO_2]$ in combination with higher temperatures will decrease regional cotton yields. Likewise Reddy and Zhao (2005); Bibi et al. (2008); Oosterhuis and Snider (2011) and Soliz et al. (2008) have shown cotton yields

to be negatively impacted by elevated temperature (direct climate change). This is in line with our findings, except for Peru and Northeast-China where we find that current yields are maintained under climate change. This is likely because present temperatures are considerably below the growth optimum and evaporative demand remains comparably low in the climate change scenarios considered here. In our model simulations, $CO_2$ fertilization overcompensates climate-change induced yield penalties and although $CO_2$ effects on cotton yields are largely unclear, FACE data support a strong positive effect (Hileman

et al., 1994; Idso and Idso, 1994; Reddy et al., 1999; Mauney et al., 1994; Mauney, 2016). Even though Kimball (2016) do not report any results on $CO_2$ fertilization effects under limited N supply, the co-limitation by nutrients is not covered here and future research should account for these effects as well (von Bloh et al., 2018). As with high, above-air temperatures (above 35 °C) the abscission of bolls increased sharply with near zero retention of bolls at 40 °C (Reddy et al., 1991; Zhao et al., 2005; Reddy et al., 2005a; Luo et al., 2014), the performance of LPJmL on cotton yields could be enhanced by introducing a yield

penalty depending on high temperature, e.g. a zero boll harvest index at temperatures above 35°C.

Cotton is a perennial, indeterminate crop and cultivated species are generally photoperiodic insensitive. Consequently, warmer temperatures will increase the length of growing season if temperature seasonality is the limiting factor (Waha et al., 2012; Minoli et al., 2019) and sufficient water and nutrients are available (Bange and Milroy, 2004; Wang et al., 2008). Hence, one major effect that reduces crop yields in annual crops (Ottman et al., 2012; Asseng et al., 2015) is not as relevant for cotton.

Climate change is associated with changes in patterns of precipitation and water availability, hence, cotton plants in some regions may be subjected to plant water deficits. Water deficit limits growth and productivity of cotton plants, and the severity of the problem may increase due to changing world climatic trends (Le Houérou, 1996). Plant water deficits depend both on the supply of water to the soil and the evaporative demand of the atmosphere. Changes in atmospheric $CO_2$ may alter the water-use efficiency of cotton production. Simultaneously, the effect of elevated $[CO_2]$ on cotton plant growth is influenced

by plant water availability. Cotton grown under $[CO_2]$ of 700 ppm use less water compared with plants grown at a $CO_2$ con-





centration of 350 ppm (Reddy et al., 1998; Ephrath et al., 2011) due to lower transpiration rates. In contrast, Samarakoon and Gifford (1995, 1996) demonstrated that cotton grown at [$CO_2$] of 710 ppm had higher plant water use than at ambient [$CO_2$] (352 ppm). FACE experiments, however, showed no differences in evapotranspiration of cotton grown at 550 ppm and ambient [$CO_2$], respectively (Dugas et al., 1994; Kimball et al., 1994; Hunsaker et al., 1994).

Our results suggest that the beneficial effects of elevated [$CO_2$] on cotton yields overcompensate yield losses from direct climate change impacts. Even though experimental evidence supports strong $CO_2$ effects on cotton and it is plausible to assume that cash crops such as cotton are grown with sufficient fertiliser applications if economically feasible, several caveats remain. First, there is only very little data on cotton grown under elevated [$CO_2$] so that the modeled response remains inherently uncertain, especially in different climate zones and at high [$CO_2$], which typically has not been investigated in experiments.

Second, negative effects of heat-days with temperatures above $35°C$ are not represented in the model. Possible negative effects on crop phenology, such as the shedding of leaves under heat and/or drought are not sufficiently understood and are also not represented in the model. Large shares of current cotton production areas are irrigated and we find that irrigated cotton production does not suffer from climate change if $CO_2$ effects are considered, whereas rainfed production is more sensitive to climate change. However, climate change also affects water availability for irrigation and has thus the potential to also

substantially affect agricultural production (Elliott et al., 2014). These effects are not considered here, as the ISIMIP protocol for agricultural production prescribes unlimited water supply for irrigation (Frieler et al., 2017). Considering these caveats, our results need to be considered optimistic. Further research on the effectiveness of long-term and high-end $CO_2$ fertilization effects as well as damages from heat is necessary to better constrain results. Accounting for constraints in freshwater availability is feasible with LPJmL in further research, yet many confounding effects, such as impacts from ozone (Schauberger et al., 2019)

or pests and diseases cannot be easily considered.

Overall, our simulation of climate change impacts on global cotton production results in similar patterns as for other crops. Given the economic relevance of cotton production in areas as Western Africa or South Asia, climate change (elevated temperature and water stress effects) poses an additional stress and deserves special attention. This holds particularly true as agriculture in these regions is already under pressure from increased demand for intensification considering rapid population

growth. Changes in virtual water content and water demands for cotton production are of special importance, as cotton production is known for its intense water consumption that led, e.g., to the loss of most of the Aral sea (Glantz, 1999; Pereira et al., 2009).

The implications of climate impacts on cotton production on the one hand, and the impact of cotton production on water resources (with major impacts particularly in India and Uzbekistan) on the other hand illustrate the need to assess how future

climate change may affect cotton production and its resource requirements. The inclusion of cotton in LPJmL allows for various large-scale studies to assess impacts of climate change on hydrological factors and its implications for agricultural production and carbon sequestration.The limited availability of data (such as valid information on tree density, irrigation management, sowing dates) substantially limits model performance and evaluation. Another issue related to data scarcity is the need for scenarios of future cropping patterns, adaptation and management as a consequence of climatic and socioeconomic change.

With climate change very likely affecting the potential growing areas of cotton (such as of other agricultural crops) and their





profitability, it is essential to provide crop yield estimates and associated water requirements under different climate scenarios to other research projects, e.g. on land-use change projections (Nelson et al., 2014). Analysing future cotton production may require more a detailed parametrisation of cotton production, allowing for the differentiation of cotton varieties, grid-cell-specific planting densities and its differentiation between irrigated and rain-fed conditions, as well as crop-specific fruit set,

which at present depends on the phenological development. The extended version of LPJmL is an important improvement as it allows for explicitly studying cotton production under climate change and associated water consumption. Results need to be carefully assessed and interpreted, as model performance remains uncertain under given constraints on data availability for model evaluation. Future work should focus on effects of climate change on irrigation water availability as well as on an implementation of heat stress effects on cotton productivity.

**5   Conclusions**

As most widely produced natural fiber cotton is of high importance to economies, but growth and irrigation water demand of cotton may be challenged by future climate change. To study how future cotton productivity is affected by projected climate change, we use the global biogeochemical model of hydrology, carbon exchange and crop growth, LPJmL, expanded to include cotton plants. Available data on observations and published estimates are used to validate the model and a set of climate

scenarios following the ISIMIP protocol to simulate global future cotton yield and water consumption. We then analyse the global cotton production and irrigation water consumption under spatially varying present and future climatic conditions. Our results suggest that the beneficial effects of elevated $[CO_2]$ on cotton yields overcompensate yield losses from direct climate change impacts, i.e. without the beneficial effect of $[CO_2]$ fertilization. While changes in atmospheric $CO_2$ may turn into enhanced water use or water-use efficiency of cotton production, the impact of elevated $CO_2$ on cotton growth depends also on

plant water availability. The extended version of LPJmL is an important improvement as it allows for explicitly studying cotton production under climate change and associated water consumption. Results need to be carefully assessed and interpreted, as model performance remains uncertain under given constraints on data availability for model evaluation. Future work should focus on effects of climate change on irrigation water availability as well as on an implementation of heat stress effects on cotton productivity.

*Code and data availability.*   The model code of LPJmL4 is publicly available through PIK's GitHub repository at https://github.com/PIK-LPJmL/LPJmL and should be cited as Schaphoff (Ed.), S.; von Bloh, W.; Thonicke, K.; Biemans, H.; Forkel, M.; Gerten, D.; Heinke, J.; Jägermeyr, J.; Müller, C.; Rolinski, S.; Waha, K.; Stehfest, E.; de Waal, L.; Heyder, U.; Gumpenberger, M.; Beringer, T.: LPJmL4 Model Code. V. 4.0. GFZ Data Services. http://doi.org/10.5880/pik.2018.002, 2018. With acceptance of this manuscript, an extended, exact version of the code and the output data from the model simulations described here will be published with DOI via GFZ Data Services under

https://doi.org/10.5880/Pik.2020.001 and should be referenced as Jans et al. (2020). In the meantime, the data can be accessed by reviewers via this review link:



http://pmd.gfz-potsdam.de/panmetaworks/review/9dc620caca61277d2f7a88d31af4908090b8a1720d722af866ae7e8d9946d7cc-pik. The review link will be deleted as soon as the DOI is active.

*Author contributions.* YJ and CM designed the study. YJ and WvB developed the model code and YJ performed the simulations. YJ prepared the manuscript with contributions from all co-authors.

*Competing interests.* The authors declare that they have no conflict of interest.

*Acknowledgements.* This study was conducted in the framework of the project 'IKI-Impact' for which we acknowledge funding from the German Ministry for the Environment (BMU,\16_II_148_Global_A_IMPACT). We thank Jens Heinke and Susanne Rolinski for valuable discussions.

*Supplement.* The supplement related to this article is available online at: https://doi.org/10.5194/hess-0-1-2020-supplement.



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
