# Peer review of "Global cotton production under climate change - Implications for yield and water consumption"

_Hydrology and Earth System Sciences, 2019_

## Referee Comment (RC1) · Anonymous Referee #1 · 11 Feb 2020

Overall, the manuscript is well-written and easy to follow. The authors' efforts on conducting a gridded cotton modeling study are commendable. However, I think some important elements, pertaining to climate change impacts, are missing. My main concern is that the authors did not account for the adverse effects of temperature rise, especially on growing season length and crop failure. This likely led to underestimation of the negative effects of climate change on yield and crop water consumption.

Specific comments:

Line 15: Please define VWC.

Line 19: Delete (and thus water stress).

Section 2: I did not see any details on model calibration. Did you have a set of parame-

ters you adjusted to match the FAO-yield? Did you follow any protocols for calibration, like changing parameters in sequence, first for yield, then for irrigation? Please explain the steps followed.

It appears that the cotton module is a new addition to the LPJmL model, has it been tested against detailed field data including in-season phenology, growth, leaf area, biomass, and yield?

Section 2: Did you apply any fertilizer? Not clear.

Line 130: Please define PFT and CFT, if not already done so.

Line 175: Why the 2090-2099 period? Typically a 30-year period is used to avoid extreme weather years, and to get average conditions. Please justify your choice. Also, the baseline period, against which the % increase is calculated, is not clear. Please specify the baseline period. Ideally, for a fair comparison, the length of years should be the same for the baseline and future periods.

Section 3.2: The authors did not mention phenology. Growing season length? How does increasing temperature affect cotton yield in your model? Shortening of growing season length is one of the major causes for yield reduction under climate change. Did you account for this?

Section 3.3: I find the term "virtual water content (VWC)" confusing. It distracts focus from the more relevant issue of "crop water consumption". To help the reader, please add text on how the changes in VWC would affect crop water consumption.

Line 283: It is worth noting, cotton yield have increased over the years in part due to the use of high yielding crop varieties as a result of breeding efforts. One of the reasons behind the temporal variation in your study?

Section 4.2: Did you account for the effects of increasing temperature on growing season length? As previously mentioned, this is important for yield and irrigation water use, and needs to be considered.

Line 313: "concluded" instead of "conclude".

Line 324: Please justify simulating climate change impacts without accounting for yield penalty due to temperature increase.

Line 327: I would be careful in stating that "warmer temperatures will increase the length of growing season". For cotton and other annual crops, increasing temperature is known to increase the rate of development, which causes the crop to mature earlier than it would at a lower temperature. Thus, this rapid development reduces the length of growing period, considering the same maturity-group cultivar is used.

Line 335-340: This section is confusing and I would caution the authors while comparing different studies. Please make sure the same matrix is compared across studies. Was it daily crop water use or seasonal; was it transpiration, evapotranspiration, or transpiration use efficiency? Please revise this section for consistency and technical accuracy.

It is expected that transpiration use efficiency would increase at higher $[CO_2]$, but the overall increase in biomass due to $[CO_2]$ enrichment, would trigger higher water consumption compared to the ambient $[CO_2]$ conditions.

Line 341: "Temperature" is another key driver of climate change impact on crop yield. Considering the crop was irrigated, rainfall likely did not have a major effect. So instead of using "direct climate change impacts", I would use "temperature rise" effects.

---

## Referee Comment (RC2) · Kokou Adambounou Amouzou (Referee) · 8 Feb 2021

The authors adapted the LPJmL model to simulate global future cotton yield and water consumption. Overall, the Manuscript is nicely prepared and well-written. The study is of high interest and very informative, providing substantial contribution for improved understanding of cotton response under projected climate change and elevated $CO_2$ in rain-fed and irrigated production systems. The authors used appropriate methods and reported well illustrated results supported by discussion and conclusions. Furthermore, the authors recognize the limitations of their study, clearly stating that the results should be treated with caution due to uncertainty in the model performance and given constraints on data availability for model evaluation. Therefore, I accept the Manuscript as is for publication in Hydrology and Earth System Sciences.

---

## Author Response (AR1)

**Reply to the editor comment on "Global cotton production under climate change – Implications for yield and water consumption" by Pieter van der Zaag**

Yvonne Jans[1,2], Werner von Bloh[1], Sibyll Schaphoff[1], and Christoph Müller[1]

[1]Potsdam Institute for Climate Impact Research, Member of the Leibniz Association, P.O. Box 60 12 03, DE-14412 Potsdam, Germany
[2]Humboldt-Universität zu Berlin, Department of Geography, Unter den Linden 6, DE-10099 Berlin, Germany

**Correspondence:** Yvonne Jans (jans@pik-potsdam.de)

*Copyright statement.* Author(s) 2020. CC BY 4.0 License.

I have one comment to add: In my opinion, the abstract and the conclusions should include the important statement in the discussion that the model outcomes "need to be considered optimistic" (lines 508-509) because of two major reasons: (1) there is generally still uncertainty about the effect of high $CO_2$ on cotton yields and water use, simply because of a lack of sufficient empirical experiments; and (2) the model used in this paper does not include the negative effects of heat-days with temperatures above 35° Celsius, which is known to lead to boll abscission and thus to lower cotton yield.

*We agree. This statement is really important because of the reasons mentioned. We followed your advice and changed the abstract (ll. 30-31) and the conclusions (ll. 460-462) accordingly.*

On a side note, I found the sentence in lines 40-43 in the abstract confusing, as it refers to "climate change effects" only, but it does not state this. Also, I found the reference to the Aral Sea in the abstract not warranted.

*Thanks for pointing to that. We revised the phrasing in the abstract (ll. 22-25) to be more clear. Additionally, we removed the reference to the Aral Sea in this section (ll. 27-28).*

There are a few typos/grammar issues that also need to be addressed (e.g. lines 157, 274, 295, 342,380).

*Thanks for pointing out the mistakes. We revised the text and corrected the spelling and grammar to our best knowledge and trust that if additional language issues should exist, these will be handled in the copy-editing process (ll. 114, 221, 242-243, 273, 278, 314).*

I look forward to receiving an improved version of the manuscript.

25   *Thank you for your valuable comments and suggestions to improve our manuscript. We hope that the revised version meets your expectations.*

**Reply to the interactive comment on "Global cotton production under climate change – Implications for yield and water consumption" by Anonymous Referee #1**

Yvonne Jans[1,2], Werner von Bloh[1], Sibyll Schaphoff[1], and Christoph Müller[1]

[1]Potsdam Institute for Climate Impact Research, Member of the Leibniz Association, P.O. Box 60 12 03, DE-14412 Potsdam, Germany

[2]Humboldt-Universität zu Berlin, Department of Geography, Unter den Linden 6, DE-10099 Berlin, Germany

**Correspondence:** Yvonne Jans (jans@pik-potsdam.de)

*Copyright statement.* Author(s) 2020. CC BY 4.0 License.

Overall, the manuscript is well-written and easy to follow. The authors' efforts on conducting a gridded cotton modeling study are commendable. However, I think some important elements, pertaining to climate change impacts, are missing. My main concern is that the authors did not account for the adverse effects of temperature rise, especially on growing season length and crop failure. This likely led to underestimation of the negative effects of climate change on yield and crop water consumption.

*Thank you for your valuable comments and suggestions to improve our manuscript and the overall positive rating of our paper. Please find our detailed response to the comments and all changes we made to the original manuscript documented in the following.*

Specific comments:

Line 15: Please define VWC.

*We acknowledge that there are different definitions for "virtual water content" (VWC) in the literature. However, we think the abstract is not the right place for providing the definition of VWC as used in this study. We therefore provide a definition of VWC when the term appears first in the main text (Section 3.1 Evaluation of model performance, ll. 226 - 228).*

Line 19: Delete (and thus water stress).

*Done as suggested.*

Section 2: I did not see any details on model calibration. Did you have a set of parameters you adjusted to match the FAO-yield? Did you follow any protocols for calibration, like changing parameters in sequence, first for yield, then for irrigation? Please explain the steps followed.

*We did not perform any yield calibration other than selecting the level of deficit irrigation that best matches observed national yield data from FAO. Typically, the best-matching irrigation level is close to yield levels at near-perfect irrigation (see whiskers in figure 1, which mostly extend to the bottom). We added information in Section 2.3, ll 91–96 and ll 202–204.*

It appears that the cotton module is a new addition to the LPJmL model, has it been tested against detailed field data including in-season phenology, growth, leaf area, biomass, and yield?

*Indeed the cotton module is a new addition, altough it is based on previous similar implementations, such as cotton for the Mediterranean region by Fader et al. (2015) or the tea implementation by Beringer et al. (2020). Following standards in global crop model evaluations (e.g. Müller et al. (2017)), we primarily focus in the model evaluation on national scale yield patterns and dynamics. However, we also compare data against data from field experiments (FACE, see figure S8 in the supplementary). We added more details in Section 2.3, ll 204–206.*

Section 2: Did you apply any fertilizer? Not clear.

*The current implementation is included in LPJmL4, a model version that does not account for nutrient limitation and fertilizers are thus not considered here. A model version that accounts for nitrogen dynamics has been developed in parallel (von Bloh et al. (2018)) and we will merge these development branches in the near future. This information is added in Section 2.2, ll 174–175 and ll 354–355.*

Line 130: Please define PFT and CFT, if not already done so.

*Plant functional types (PFT) and crop functional types (CFT) were defined in ll 70–72 (section 2.1 The LPJmL model).*

Line 175: Why the 2090-2099 period? Typically a 30-year period is used to avoid extreme weather years, and to get average conditions. Please justify your choice. Also, the baseline period, against which the % increase is calculated, is not clear. Please specify the baseline period. Ideally, for a fair comparison, the length of years should be the same for the baseline and future periods.

*Thanks for poiting to that. We will calculate future yields for the period 2070-2099 and compare the values to the baseline period 1980-2009 in the revised manuscript.*

Section 3.2: The authors did not mention phenology. Growing season length? How does increasing temperature affect cotton yield in your model? Shortening of growing season length is one of the major causes for yield reduction under climate change. Did you account for this?

*The cotton plant is a perennial plant and the warming-driven acceleration of phenological development thus plays out differently than in annual agricultural plants, such as wheat and rice, where indeed much of the climate change damage can be attributed to the shortened growing season, if no adaptation measures are taken to counteract that (e.g. choosing slower-maturing cultivars). We now provided additional information on phenology and growth in Section 2.2, ll 107–111, ll 114–117, ll 118–120, in Section 2.3, ll 199–201 and extended Section 3.2, ll 262–265 accordingly.*

Section 3.3: I find the term "virtual water content (VWC)" confusing. It distracts focus from the more relevant issue of "crop water consumption". To help the reader, please add text on how the changes in VWC would affect crop water consumption.

*Indeed, the reduced VWC comes from increased productivity that outweighs the slightly increased global cotton water consumption. We have added that information in Section 3.3, ll 285–286 and ll 297–300.*

Line 283: It is worth noting, cotton yield have increased over the years in part due to the use of high yielding crop varieties as a result of breeding efforts. One of the reasons behind the temporal variation in your study?

*We added a short discussion of the role of high-yielding varieties to observed and projected yield trends in Section 4.1, ll 314–316.*

Section 4.2: Did you account for the effects of increasing temperature on growing season length? As previously mentioned, this is important for yield and irrigation water use, and needs to be considered.

*With cotton as a perennial plant, increasing temperatures do not affect growing season length in the same way as they do in annual crops (e.g. wheat, rice). Please see above for the answer to the similar question above and corresponding changes in Section 2.2, ll 107–111, Section 2.3, ll 199–201. We have also extended the discussion of this point in section 4.2, ll 362-365.*

Line 313: "concluded" instead of "conclude".

90     *Thanks for the catch. Done.*

Line 324: Please justify simulating climate change impacts without accounting for yield penalty due to temperature increase.

*This is a misunderstanding. The model does account for yield responses due to temperature increase, which can lead to*
95 *declining or increasing yields, depending on the starting temperature and the extent of temperature increase. What we meant to say here is that we do not yet account for an explicit damage mechanism to extreme heat events, where e.g. an abscission of cotton balls can occur. We have clarified this in Section 4.2, ll 358–361.*

Line 327: I would be careful in stating that "warmer temperatures will increase the length of growing season". For cotton
100 and other annual crops, increasing temperature is known to increase the rate of development, which causes the crop to mature earlier than it would at a lower temperature. Thus, this rapid development reduces the length of growing period, considering the same maturity-group cultivar is used.

*This is true, but cotton is not an annual but a perennial indeterminate plant, even though it is predominantly managed as*
105 *an annual crop (i.e. sown and removed each year). This is exactly the point we make here: the reduction in growing season length from accelerated phenology, as identified as a major mechanism for declining yields in annual crops does not apply here in the same way. We revised the phrasing in Section 4.2, line 363 and added information in ll 367–368 to make this clearer.*

Line 335-340: This section is confusing and I would caution the authors while comparing different studies. Please make sure
110 the same matrix is compared across studies. Was it daily crop water use or seasonal; was it transpiration, evapotranspiration, or transpiration use efficiency? Please revise this section for consistency and technical accuracy.

It is expected that transpiration use efficiency would increase at higher $[CO_2]$, but the overall increase in biomass due to $[CO_2]$ enrichment, would trigger higher water consumption compared to the ambient $[CO_2]$ conditions.
115
*Thanks for pointing to this ambiguity. We agree that the text should state more clearly that water use efficiency and water consumption do not necessarily change simulateneously or similarly. We try to discuss effects at the level of general mechanisms, indicating that crop water consumption under elevated $[CO_2]$ can also be expected to respond to water use efficiency, leaf biomass and water availablity. We revised this part accordingly, ll 375–381.*
120
Line 341: "Temperature" is another key driver of climate change impact on crop yield. Considering the crop was irrigated, rainfall likely did not have a major effect. So instead of using "direct climate change impacts", I would use "temperature rise" effects.

125 *We agree. Indeed, temperature is another key driver of climate change impact on crop yield. However, since large areas of cotton production are not irrigated (compare Figure S2b), we keep using the term "direct climate change impacts" here, but add (temperature rise, changes in precipitation) to be more specific.*

**References**

130    Beringer, T., Kulak, M., Müller, C., Schaphoff, S., and Jans, Y.: First process-based simulations of climate change impacts on global tea production indicate large effects in the World's major producer countries, Environmental Research Letters, 15, 034 023, 2020.

Fader, M., von Bloh, W., Shi, S., Bondeau, A., and Cramer, W.: Modelling Mediterranean agro-ecosystems by including agricultural trees in the LPJmL model, Geoscientific Model Development, 8, 3545–3561, https://doi.org/10.5194/gmd-8-3545-2015, https://www. geosci-model-dev.net/8/3545/2015/, 2015.

135    Müller, C., Elliott, J., Chryssanthacopoulos, J., Arneth, A., Balkovic, J., Ciais, P., Deryng, D., Folberth, C., Glotter, M., Hoek, S., et al.: Global gridded crop model evaluation: benchmarking, skills, deficiencies and implications, Geoscientific Model Development, 10, 1403–1422, 2017.

von Bloh, W., Schaphoff, S., Müller, C., Rolinski, S., Waha, K., and Zaehle, S.: Implementing the Nitrogen cycle into the dynamic global vegetation, hydrology and crop growth model LPJmL (version 5.0), Geoscientific Model Development, 11, 2789–2812,

140    https://doi.org/10.5194/gmd-11-2789-2018, https://www.geosci-model-dev.net/gmd-11-2789-2018/, 2018.

**Reply to the interactive comment on "Global cotton production under climate change – Implications for yield and water consumption" by Kokou Adambounou Amouzou (Referee #2)**

Yvonne Jans[1,2], Werner von Bloh[1], Sibyll Schaphoff[1], and Christoph Müller[1]

[1]Potsdam Institute for Climate Impact Research, Member of the Leibniz Association, P.O. Box 60 12 03, DE-14412 Potsdam, Germany
[2]Humboldt-Universität zu Berlin, Department of Geography, Unter den Linden 6, DE-10099 Berlin, Germany

**Correspondence:** Yvonne Jans (jans@pik-potsdam.de)

*Copyright statement.* Author(s) 2020. CC BY 4.0 License.

The authors adapted the LPJmL model to simulate global future cotton yield and water consumption. Overall, the manuscript is nicely prepared and well-written. The study is of high interest and very informative, providing substantial contribution for improved understanding of cotton response under projected climate change and elevated $CO_2$ in rain-fed and irrigated production systems. The authors used appropriate methods and reported well illustrated results supported by discussion and conclusions. Furthermore, the authors recognize the limitations of their study, clearly stating that the results should be treated with caution due to uncertainty in the model performance and given constraints on data availability for model evaluation. Therefore, I accept the manuscript as is for publication in Hydrology and Earth System Sciences.

*Thank you for your time to evaluate our paper and your recommendation to accept it for publication!*